# Inclusive Tourism Adopted to Geosites: A Study in the Ajodhya Hills of West Bengal in India

Avijit Ghosh [1], Rahul Mandal [2] and Premangshu Chakrabarty [1,*]

1 Department of Geography, Visva-Bharati University, Bolpur 731235, WB, India; avijitgh042@gmail.com
2 Rasa Rajlakshmi High School, Khayrasole 731125, WB, India; rahulskbu1992@gmail.com
* Correspondence: drpremangshuindia@gmail.com

**Abstract:** Inclusive tourism is a specialized branch of tourism emphasizing the inclusion of the disabled, who otherwise could not participate in tourism, despite having wealth to spend for leisure and recreation. There exists a research gap in analyzing scope of geotourism in this context. Disabilities affecting access to geosites affect geotourism since most of the geosites all over the world are situated in difficult terrain from the stand point of accessibility. It is inclusive tourism, also called accessible tourism, that facilities the consumers to reach the desired destinations. The present study assesses such destinations in Ajodhya hills, located in West Bengal, India, a geotourist's paradise in terms of rarity and diversity, aesthetic appeal, and cultural value. The study derives an accessibility–attraction model to identify inclusive tourism planning priorities from tourism marketing perspectives. Extensive field work followed by the application of qualitative methods of data analysis yield results dedicated to sustainable geotourism development. The discussion reveals the scope of developing specific facilities, using GIS, which encourage physically challenged people to visit geosites and simultaneously fulfil the objective of guiding planners and policy makers to identify and develop more suitable sites for introducing inclusive tourism facilities.

**Keywords:** disabilities; accessible; destination; geotourist; sustainable





## 1. Introduction

The fundamentals of inclusive tourism development consist of the inclusion of people with disabilities in society, fulfilling the leisure and recreation needs of differently abled and elderly tourists. Sustainable Development Goals in tourism could not be achieved without the inclusion of people with disabilities, who are a historically excluded and socially marginalized group who lack power and voice [1,2]. Such people are often excluded from various types of leisure activities due to the barrier-laden and socially exclusive nature of the activities in question [3]. Inclusive tourism is designed to enhance the quality of life and satisfaction of people with disabilities. As, with enhanced accessibility, they could enjoy various tourism products, they are beneficiaries of inclusive tourism [4]. Inclusive tourism is a rather post-modern concept aiming to achieve sustainability with a focus on the competitiveness of the tourism industry, and catering, in general, to the needs of mobility impaired tourists and, specifically, elderly tourists. Inclusive tourism is thereby able to add value to tourism and contribute to knowledge and understanding by seeking to explicitly overcome the exclusionary tendencies of tourism [5]. Such tourism is characterized by the development of various products and services dedicated to disabled people irrespective of their gender, age, or physical status, providing them accessibility to attractions, for which it is also called "accessible tourism" [6].

Accessible tourism, therefore, is based on advocating for the rights of differently abled people to enjoy holidays and tourism infrastructure, which necessitates removing barriers [7]. The basic purpose of such infrastructure is to provide safe and comfortable tourism experiences for all holidaymakers, regardless of physical/cultural disabilities

relating to gender, race etc. [8]. It is also called "disabled tourism" in the sense that elderly travelers, pregnant women, parents pushing their children in strollers, or even people with temporary injuries, may enjoy better tourism experiences if they opted for inclusive tourism packages in their holidays and leisure time [9,10]. The inclusive tourism network provides various kinds of services and facilities that make tourist destinations accessible to all people, regardless of their physical limitations, disabilities, or age. This subcategory of tourism has thus evolved as a tool to execute the social right to cater to people with special needs in a more holistic way [11]. Ideologically it is conceived that a handicap might be God-given but a disability is often man-made because of failures in existing facilities [12]. The aim of promoting inclusive tourism is to provide a horizon of diversified facilities and services in an equitable, non-discriminatory manner [13]. This paper is an attempt to analyze the aspects of the development of inclusive tourism in the Ajodhya hills of Purulia (Figure 1), which are famous for their geotourism resources. The following objectives have been undertaken for present study:

- To review the rarity and diversity aspects, aesthetic appeal, and cultural values of hill-based and water-based geotourism sites in the Ajodhya hills;
- To enumerate the status of attractions and accessibilities of major geotourism sites of the Ajodhya hills from an inclusive tourism perspective;
- To evaluate the scope of planning to facilitate differently abled and elderly tourists in moderately and less accessible areas with highly attractive geotourism sites.

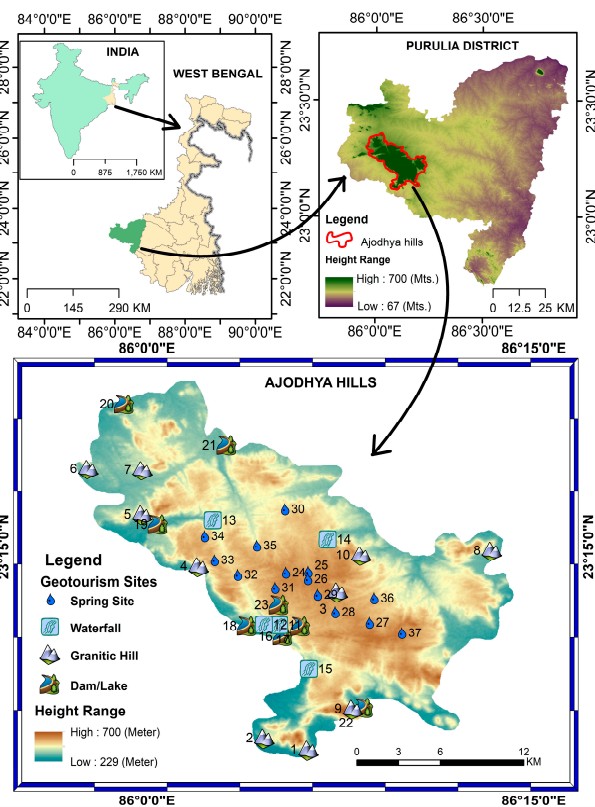

**Figure 1.** Location map of the specific geotourism sites in theAjodhya hills area using ArcGIS, version 10.5 (1. Pakhi Pahar; 2. Mathaburu; 3. Mayur Pahar; 4. Kuhuburu; 5. Chemtaburu; 6. Jajahatu; 7. Ganjaburu; 8. Gojaburu; 9. Gorgaburu; 10. Chandni Pahar; 11. Bamni falls; 12. Turga falls; 13. Machkanda falls; 14. Ghageswari falls; 15. Dauri khal; 16. Lower dam; 17. Upper dam; 18. Turga lake; 19. Khairabera lake; 20. Norahara dam; 21. Muruguma lake; 22. Pardi lake; 23. Marble lake; 24. Bhuigara spring; 25. Shimulghutu spring; 26. Dharkuli spring; 27. Puniasasan spring; 28. Spring Sitakunda; 29. Andhra spring; 30. Chunkati spring; 31. Ranga spring; 32. Saldih spring; 33. Kurpahar spring; 34. Jilingara spring; 35. Bhonsodi spring; 36. Chatni spring; 37. Edelbera spring).

## 2. Study Area

The Ajodhya hills is basically the easternmost extension of the Chotanagpur plateau (height 300–665 m), which is located in the south-western part of the Purulia district of West Bengal. Geologically, it is a Pre-Cambrian metamorphic terrain composed of granite andgneissic rocks of the "Chotanagpur Granite Gneissic Complex" (CGGC), which originated from tectonic activities roughly more than 1.6 billion years ago [14]. The topographic expressions bear the evidence of multi-cyclic tectonic events, magmatic activities, and metamorphism [15,16].

The shape of the Ajodhya hills is like the English capital letter "L", and it covers an area of almost 400 sq. km. By virtue of a complex evolutionary history, both multi-cyclic and polygenic landforms are abundant [17]. Residual hills, escarpments, river valleys, waterfalls, lakes, springs, etc., are among its valuable geosites that attract the attention of geotourists. From a geotourism perspective, all its geomorphic divisions are significant, namely:

- The plateau top, which is almost flat with isolated peaks and hillocks;
- The plateau rim, which is also called the break-of-slope zone;
- The piedmont zone at the foothill.

Geosites, which are being identified worldwide through various schemes of classification adopted by various scholars [18–20], are nothing but the relief forms on the Earth's surface with scientific, aesthetic, ecological, cultural, and economic values. To be very specific, a geosite, as a landform of geological heritage [21,22], represents the particular aspects of relief determined by morphogenetic processes and the geographic sublayer [23]. The thirty-seven sites (Figure 1) of geological interests scattered in the aforementioned geomorphic divisions comprise such qualified geosites in accordance with various classification schemes applied in different geographic environments [24–27]. In promoting inclusive tourism in such geosites, we raise the scope of appreciation of their rarity and diversity in terms of their scientific, educational, aesthetic, cultural, and recreational values, serving the interest of current as well as future generations [28]. It has been observed that during visits to sites of geological interest, the guests are naturally keen in understanding the geomorphological treasures [17]. If inclusive tourism is introduced, the influx of visitors will increase, and more employment opportunities will arise by which geotourism will surely serve the nation. However, simultaneously, to avoid the evils of mass tourism growth, it is essential to remain alert to heritage protection aspects so that inclusive tourism may become synonymous with responsible tourism. Being a mosaic of geological entities of special scientific importance, these geosites and geomorphosites are representative of the region's geological history and of the events and processes that shaped it [29]. The study focuses on various forms of these sites in the study region (classifying them into rock-based and water-based sites), enumerating the status of their attraction and accessibility to serve the needs of sustainable planning in order to promote inclusive tourism.

## 3. Materials and Methods

The study begins with a literature triangulation on the geology and geomorphology of the Ajodhya hills to extract the significance of its landscape features which are attracting geotourists, with a vivid focus on their rarity and diversity aspects. The specific locations of these geosites have been identified using GPS and photographed during field visits. During the field survey, the cultural values of the sites relating to the beliefscape of indigenous people residing in the region have also been registered. Further, differently abled and elderly tourists visiting the Ajodhya hills have been randomly interviewed during tourist season (2022–2023), as well as during the annual hunting festival of indigenous people. The photographs of geosites have been displayed in order to obtain quantitative data on the aesthetic appeal of the sites. Utilizing the database developed during the field work, the inclusive tourism status of the existing geosites has been enumerated, applying equal weightage to each and every sub-criterion. Since three sub-criteria (Table 1) considered for both accessibility and attraction, a 3-point scale is applied for deriving data from visitors

to perform an attraction-accessibility analysis; the maximum score is 3 and the minimum score is 1.

**Table 1.** Attraction-accessibility interface for geosite assessment in the context of inclusive (accessible) tourism.

| Criteria | Sub-Criteria | Three Point Rating Scale | | |
|---|---|---|---|---|
| | | 1 | 2 | 3 |
| Accessibility | Access of vehicle to geotourism attraction point | >1 km from nearby road | <1 km, but more than 500 mts. | <500 mts. |
| | Access path to geotourism attraction point | Steep Staircase/No stair | Gentle Staircase/Foot path | Tourist tract suitable for wheel chair access |
| | Access to geotourism attraction point | Difficult slope (Moderate to steep i.e., >10 to 25 degree) | Gentle to Moderate Slope (>2 to 10 degree) | Level Slope (0–2 degree) |
| Attraction | Rarity and Diversity | Ordinary | Conspicuous | Unique |
| | Aesthetic appeal | Low | Medium | High |
| | Cultural value | Negligible | Significant for association with belief system | Outstanding for mythology/ folklore |

Source: Model developed by the authors.

The discrimination of geosites of the Ajodhya hills based on accessibility and attraction is the output of this model, based on which planning priorities have been identified. Using QGIS software version 3.22, a planning map with inclusive tourism facilities has been developed, while the location map displaying geotourism sites has been prepared using ArcGIS, version 10.5.

## 4. Results and Discussion

Major geotourism attractions of the Ajodhya hills, which are an extension of the geologically famous Chotanagpur plateau region of India, could be classified into the following groups.

### 4.1. Major Hills as Geosites

Hill sites (Figure 2) are one of the major attractive geosites of the Ajodhya hills. Two hill sites (called Pahar) are very popular, namely, the Pakhi Pahar and Mayur Pahar. The word *Pakhi* means bird, numerous species of which are painted on its rock face via the effort of an NGO involving artists. These pictographs have added enormous cultural value in tourists' perception. The spectacular conical view of Pakhi hill, the marvelous block disintegration site near it, the elongated rock outcrop, and the natural scenic view from the Mayur hill are examples of hill-based geotourism sites that draw huge amount of tourists. Both the hills are connected with metal roads from different directions for which these two sites are already popular.

Chemtaburu and Gojaburu are, however, considered as potentially paradigmatic geotourism sites of the Ajodhya hills for the problem of less connectivity. Chemtaburu is the highest peak of Ajodhya hill complex. According to local dialect, *Chemta* means flat and the word *Buru* means hill. Chemtaburu, made of granite and gneissic rocks, looks like a crocodile's back and is very steep at the margin, whereas Gojaburu manifests itself as a conical shaped structure of the same rocks. These two lofty and steep granitic hills are popular for trekking or hill climbing but inaccessible, due to steepness, to the old or disabled tourists. Nonetheless, they have tremendous geomorphic and aesthetic appeal. Table 2 provides the status of hill-based sites from a geotourism perspective.

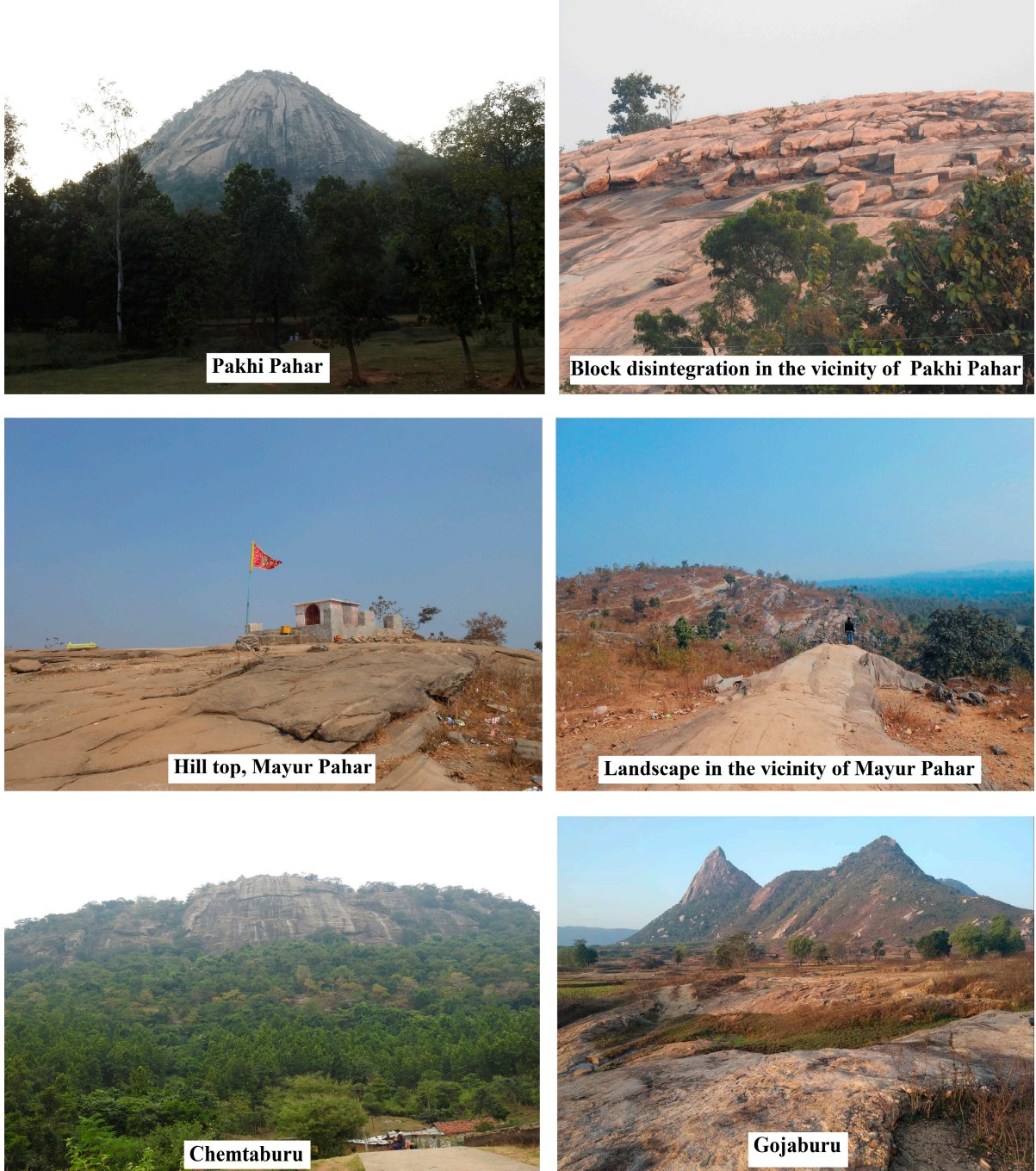

**Figure 2.** Hill sites and spectacular geomorphic features in Ajodhya hills.

**Table 2.** Hill-based geosites of Ajodhya hills, Purulia.

| Sl. No. | Name of Hills | Height (m) | Rarity & Diversity | Aesthetic Appeal | Cultural Values |
|---|---|---|---|---|---|
| 1 | Gorgaburu | 642 | Ordinary | Medium | Significant: association with belief system |
| 2 | Pakhi Pahar | 480 | Unique | High | Outstanding: known for rock art painted over it |
| 3 | Mathaburu | 478 | Conspicuous | Medium | Outstanding: for mythology/folklore |
| 4 | Chemtaburu | 700 | Unique | High | Significant: association with belief system |
| 5 | Ganjaburu | 602 | Conspicuous | Medium | Outstanding: for mythology/folklore |
| 6 | Jajahatu | 575 | Conspicuous | Medium | Negligible |
| 7 | Kuhuburu | 468 | Conspicuous | Medium | Outstanding: for mythology/folklore |
| 8 | Gojaburu | 585 | Conspicuous | High | Outstanding: for mythology/folklore |
| 9 | Chandni Pahar | 620 | Ordinary | Low | Negligible |
| 10 | Mayur Pahar | 598 | Conspicuous | High | Significant: association with belief system |

Source: Field work, 2022–2023.

## 4.2. Waterbodies with Attractive Rock Exposures

Among the lakes, Marble Lake is unique because of the exposure of spectacular geological features (Figure 3), while other conspicuous waterbodies also exert aesthetic appeal for the geotourists (Table 3).

**Table 3.** List of dams/lakes in Ajodhya hills and its attraction values.

| Sl. No. | Name | Area (Approx.) in sq.km. | Rarity & Diversity | Aesthetic Appeal | Cultural Values |
|---|---|---|---|---|---|
| 1 | Muruguma lake | 0.80 | Conspicuous | High | Negligible |
| 2 | Norahara dam | 0.03 | Ordinary | Low | Negligible |
| 3 | Khairabera lake | 0.38 | Conspicuous | High | Negligible |
| 4 | Turga dam | 0.19 | Ordinary | High | Negligible |
| 5 | PPSP Upper dam | 0.98 | Conspicuous | Medium | Negligible |
| 6 | PPSP Lower dam | 0.32 | Conspicuous | High | Negligible |
| 7 | Durgabera dam (Marble lake) | 0.02 | Unique (spectacular geological features to see) | High | Negligible |
| 8 | Pardi lake | 0.07 | Conspicuous | High | Negligible |

Source: Field survey, 2022–2023.

Marble Lake was originally an abandoned mine site on the way to Bamni falls. A stone quarry utilized for building PPSP (Purulia Pump Storage Project) wasturned into an artificial water body and named Marble Lake. Among the features around it that attract geotourists, the most conspicuous is "colour banding", i.e., melanosome (dark colour zone in rock) and palaeosome (light colour zone in rock). This type of excellent geological feature arose due to the migmatization of the parent rock in ultra-high temperatures (UHT) [30]. The dolerite dykes and sills and pegmatite veins intruded during post-metamorphism and deformation, while the hinges of the horizontal folds are disrupted by the intrusion [31]. Excellent cross-cutting relationship can be observed between pegmatite veins and dolerite dykes where pegmatite veins are cut by the younger dolerite dykes [16]. The quartz veins are often boudinaged, resulting into another mesmerizing feature, namely, "Pinch and Swell" structures, in which the width of the quartz veins alternatively increase and decrease. It is worth mentioning that the pegmatite and quartz veins occurred due to the invasion of hydrothermal fluid, displaying coarse crystals of quartz, feldspar, and mica minerals.

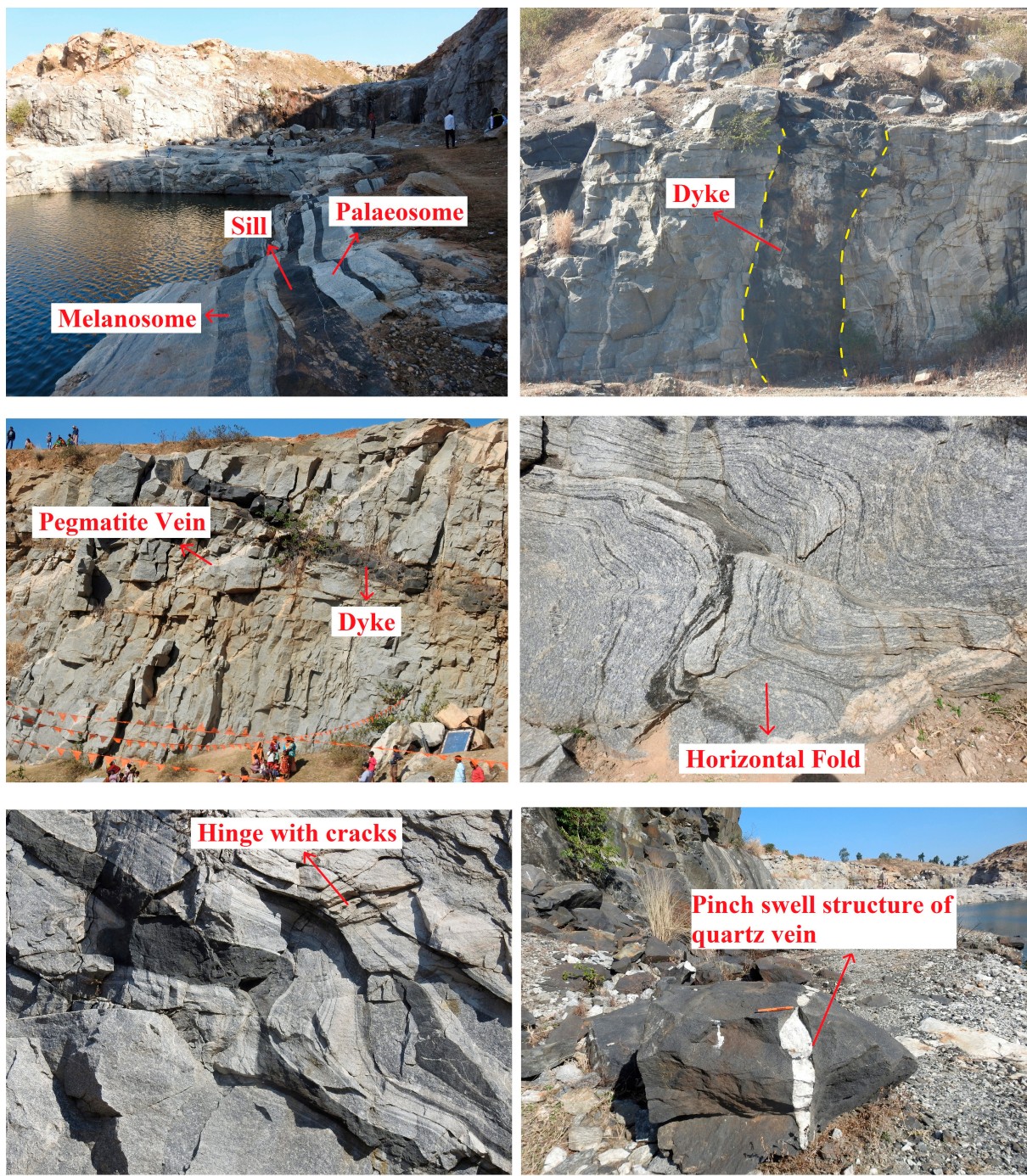

**Figure 3.** Marvelous geological features around the marble lake.

### 4.3. Major Waterfall Sites Drawing Geotourists

The dimensions of waterfalls along the line of break of slopes depend on nature of terrain, lithological as well as structural characteristics, and channel flow [32,33]. In the Ajodhya hills, the waterfalls (Table 4, Figure 4) are characteristics features of the scarp land of the plateau margin, serving as the source of many small and large streams.

**Table 4.** List of waterfalls in Ajodhya hills and its attraction values.

| Sl. No. | Name | Approx. Height (in meter) | Approx. Width (in meter) | Rarity & Diversity | Aesthetic Appeal | Cultural Value |
|---|---|---|---|---|---|---|
| 1 | Bamni Waterfall (Cascading/Horse tail) | 52 | 11 | Unique | High | Negligible |
| 2 | Turga Waterfall (Cascading) | 22 | 10.5 | Conspicuous | High | Negligible |
| 3 | Machkanda Waterfall (Cascading/Horse tail) | 50 | 6 | Unique | High | Negligible |
| 4 | Ghageswari Waterfall (Cascading) | 26 | 5 | Conspicuous | High | Negligible |
| 5 | Dauri khal (Rapid) | 24 | 6 | Unique | High | Negligible |

Source: Field survey, 2022–2023.

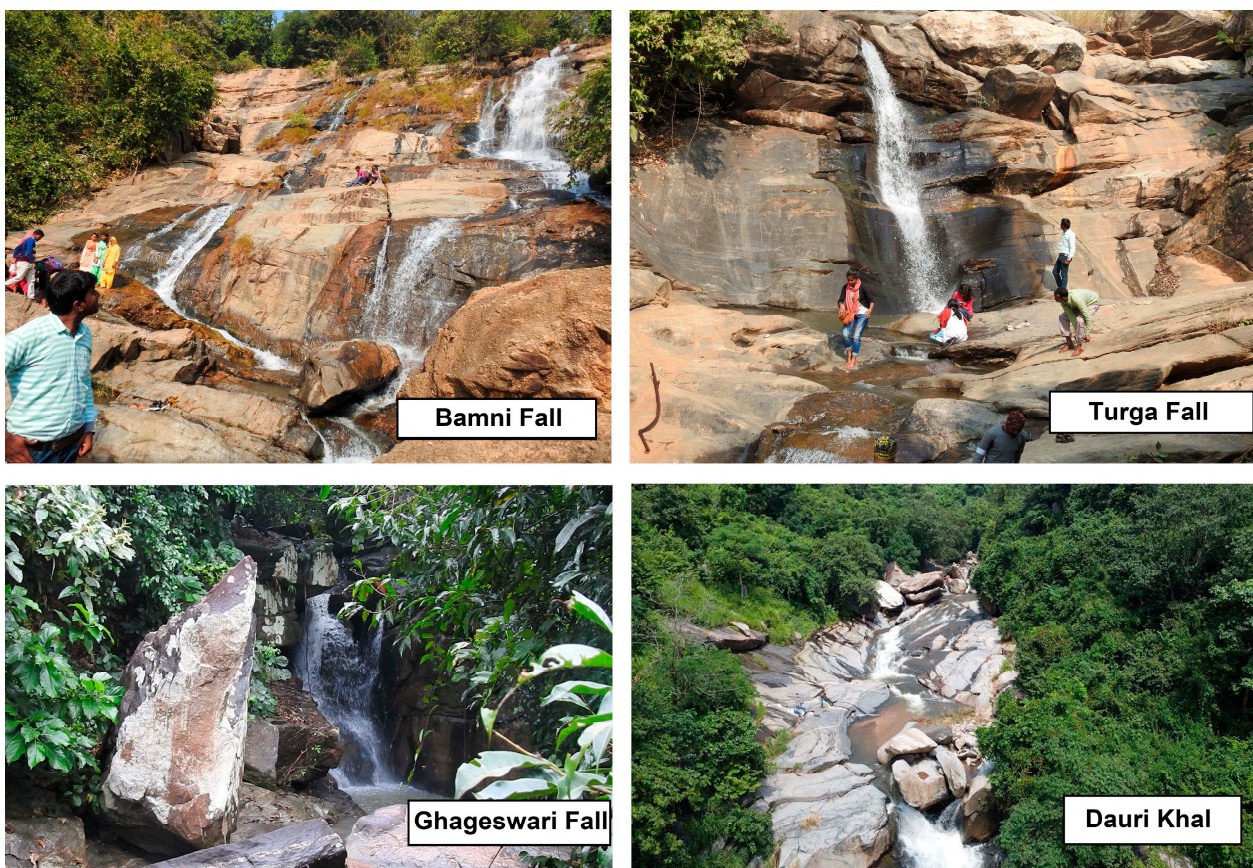

**Figure 4.** Picturesque waterfalls of Ajodhya hills.

Among the falls, Bamni, Turga and Dauri khal are situated at the southern plateau fringe, while the Machkanda falls and Ghageswari falls lie at the northwestern and northeastern part of the landscape. Bamni and Machkanda are unique waterfalls, as revealed from visitors' response; however, due to accessibility bottlenecks, Machkanda is less visited while Bamni and Turga are popular geotourism sites of Ajodhya hills.

### 4.4. Springs of Ajodhya Hills as Geotourism Attraction

A spring line is very much prominent (Figure 5) in the case of the Ajodhya hills because of the presence of the confined aquifer mostly between a granite–gneissic parent material at the bottom and reddish/brownish red soil layers on the top. They act as the source of many rivulets (Table 5), such as Rupai, Karru, Kumari, Hanumatya, Kulbera, Saharjor, Chunmatia, Hensadi, Turga, Kistobazar Nala, etc. [34]. Sitakunda is one of the popular spring sites of Ajodhya hills, which bears immense cultural value. According to mythological belief, during their exile, Lord Rama, the hero of the Great Indian Epic Ramayana, pierced an arrow through the Earth's crust for his thirsty wife Sita and the pristine water came out. Due to this legend, the spring is considered sacred; therefore, it is attractive to the domestic visitors. Due to better accessibility at the site, it is already popular. Other spring sites, such as Kurpahar, Bhuigara, and Dharkuli, are also easily accessible, but the more attractive Bhonsodi and Saldih need the development of a safe access path for geotourists.

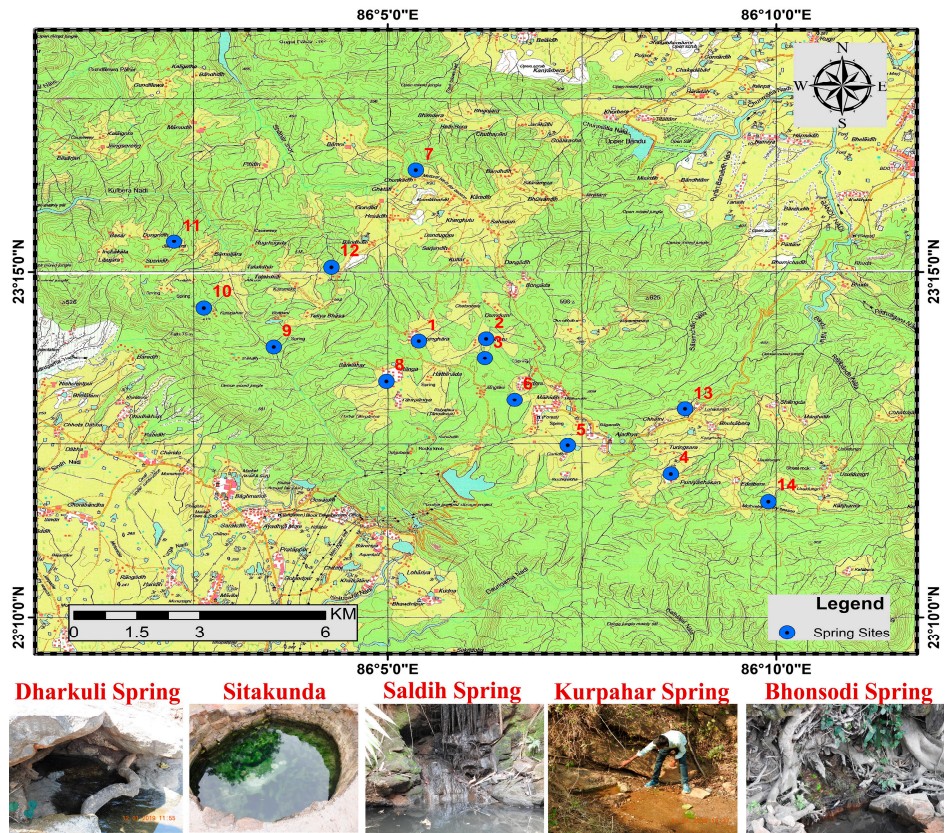

**Figure 5.** Spring line of Ajodhya hills (1. Bhuigar; 2. Shimulghutu; 3. Dharkuli; 4. Puniasasan; 5. Sitakunda; 6. Andhra; 7. Chunkati; 8. Ranga; 9. Saldih; 10. Kurpahar; 11. Jilingara; 12. Bhonsodi; 13. Chatni; 14. Edelbera). Source: SOI toposheet 73 I/3 and 73 I/4 through Arc GIS 10.5 to represent area of interest.

**Table 5.** List of springs in the Ajodhya hills and attraction values.

| Sl. No. | Name of the Spring | Stream Originated | Height From msl (mts.) | Rarity & Diversity | Aesthetic Appeal | Cultural Value |
|---|---|---|---|---|---|---|
| 1 | Bhuigara | Hensadi | 578 | Unique | High | Significant: association with belief system |
| 2 | Shimulghutu | Sahar Jhor | 552 | Ordinary | Low | Negligible |
| 3 | Dharkuli | Sahar Jhor | 585 | Unique | High | Significant: association with belief system |
| 4 | Puniasasan | Bandu | 548 | Conspicuous | Medium | Significant: association with belief system |
| 5 | Sitakunda | Kistobazar | 425 | Unique | High | Strong mythological connection |
| 6 | Andhra | Kistobazar | 590 | Ordinary | Low | Negligible |
| 7 | Chunkati | Chunmatia Nala | 500 | Conspicuous | Medium | Negligible |
| 8 | Ranga | Turga | 566 | Ordinary | Low | Negligible |
| 9 | Saldih | Saldih | 525 | Unique | High | Negligible |
| 10 | Kurpahar | Kulbera | 511 | Conspicuous | High | Negligible |
| 11 | Jilingara | Kulbera | 508 | Ordinary | Low | Negligible |
| 12 | Bhonsodi | Sahar Jhor | 522 | Unique | High | Negligible |
| 13 | Chatni | Unnamed Stream | 502 | Ordinary | Low | Negligible |
| 14 | Edelbera | Kumari River | 582 | Ordinary | Medium | Negligible |

Source: Field survey, 2022–2023.

With the application of the attraction accessibility model developed by the authors for the available geosites (Table 6), nine site categories have been identified (Figure 6):

i. Low attraction, High accessibility (LH): Unsuitable for inclusive tourism;
ii. Low attraction, Medium accessibility (LM): Unsuitable for inclusive tourism;
iii. Low attraction, Low accessibility (LL): Unsuitable for inclusive tourism planning;
iv. Medium attraction, High accessibility (MH): Provision of infrastructure may yield good results;
v. Medium attraction, Medium accessibility (MM): Unsuitable for investment;
vi. Medium attraction, Low accessibility (ML): Unsuitable for investment;
vii. High attraction, High accessibility (HH): Conducive to inclusive tourism;
viii. High attraction, Medium accessibility (HM): Suitable for the immediate development of inclusive tourism;
ix. High attraction, Low accessibility (HL): Huge investment required for inclusive tourism development.

**Table 6.** Analysis of geosites following the attraction-accessibility model.

| Sl. No. | Geosite Name | Attraction (Mean) | Accessibility (Mean) | Category (***) |
|---|---|---|---|---|
| 1 | Pakhi Pahar | 3.00 | 3.00 | HH |
| 2 | Mathaburu | 2.33 | 1.67 | HM |
| 3 | MayurPahar | 2.33 | 2.33 | HH |
| 4 | Kuhuburu | 2.0 | 1.33 | ML |
| 5 | Chemtaburu | 2.67 | 1.33 | HL |
| 6 | Jajahatu | 1.67 | 2.00 | MM |
| 7 | Ganjaburu | 2.00 | 1.00 | ML |
| 8 | Gojaburu | 2.67 | 1.67 | HM |
| 9 | Gorgaburu | 1.67 | 1.67 | MM |
| 10 | Chandni Pahar | 1.00 | 1.33 | LL |
| 11 | Bamni falls | 2.33 | 1.33 | HL |
| 12 | Turga falls | 2.00 | 2.00 | MM |
| 13 | Machkanda falls | 2.33 | 1.33 | HL |
| 14 | Ghageswari falls | 2.00 | 1.67 | MM |
| 15 | Dauri khal | 2.33 | 1.00 | ML |

**Table 6.** *Cont.*

| Sl. No. | Geosite Name | Attraction (Mean) | Accessibility (Mean) | Category (***) |
|---|---|---|---|---|
| 16 | Lower dam | 2.00 | 3.00 | MH |
| 17 | Upper dam | 1.67 | 3.00 | MH |
| 18 | Turga dam | 1.33 | 2.67 | LH |
| 19 | Khairabera dam | 2.00 | 3.00 | MH |
| 20 | Norahara dam | 1.00 | 2.33 | LH |
| 21 | Muruguma dam | 2.00 | 3.00 | MH |
| 22 | Pardi Lake | 2.00 | 3.00 | MH |
| 23 | Marble Lake | 2.33 | 2.33 | HH |
| 24 | Bhuigara | 2.67 | 2.67 | HH |
| 25 | Shimulghutu | 1.00 | 1.67 | LM |
| 26 | Dharkuli | 2.67 | 2.67 | HH |
| 27 | Puniasasan | 2.00 | 2.67 | MH |
| 28 | Sitakunda | 3.00 | 2.67 | HH |
| 29 | Andhra | 1.00 | 2.67 | LH |
| 30 | Chunkati | 1.67 | 2.67 | MH |
| 31 | Ranga | 1.00 | 2.67 | LH |
| 32 | Saldih | 2.33 | 1.33 | HL |
| 33 | Kurpahar | 2.00 | 2.33 | MH |
| 34 | Jilingara | 1.00 | 2.00 | LM |
| 35 | Bhonsodi | 2.33 | 1.33 | HL |
| 36 | Chatni | 1.00 | 1.33 | LL |
| 37 | Edelbera | 1.33 | 1.67 | LM |

Source: Computed by the authors based on perception responses of target group visitors. *** All border cases are located in the higher category in view of expectations of improvement in the near future.

Among the geosites (Table 6), the HM and HL group of sites could be chosen for inclusive tourism planning for the first phase because their higher level attractivities could draw more geotourists from the differently abled/elderly community if appropriate planning measures were to be undertaken. Figure 7 represents the scope of introducing inclusive tourism facilities in this context. Two ropeways, one from Chemtaburu to Machkanda falls (5.5 km long) and another from Dauri khal to Bamni falls (3.85 km long), are essential to developing inclusive tourism in the area. To provide access to the base of the falls, elevators could be provided. Self-help groups of women are already engaged in Pakhi Pahar [35], who may be further trained to serve the differently abled and older tourists and enable them to enjoy the rock art for which the hill is famous. *Doolie* (palanquin)/wheel chairs operated by them could draw the old/disabled into geotourism. With its increasing popularity, a sky walk with an elevator could also be planned if funds permit. Descriptions of geosites by guides trained specifically to serve inclusive tourists who simultaneously need physical help are an immediate necessity.

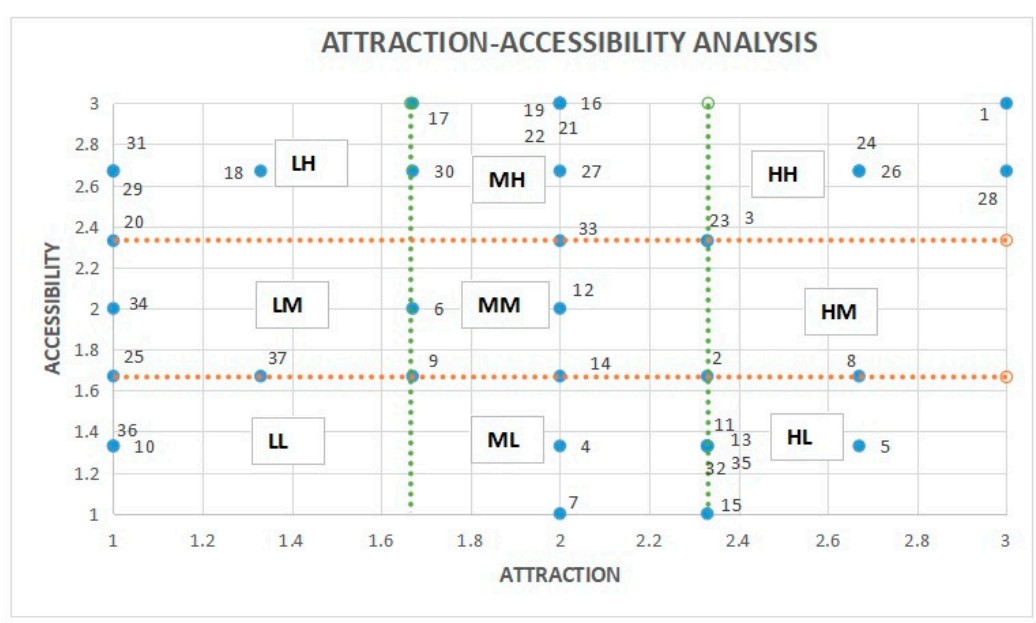

**Figure 6.** Graph showing status of each geosite in context of inclusive tourism development. Source: Field data analysis through MS EXCEL.

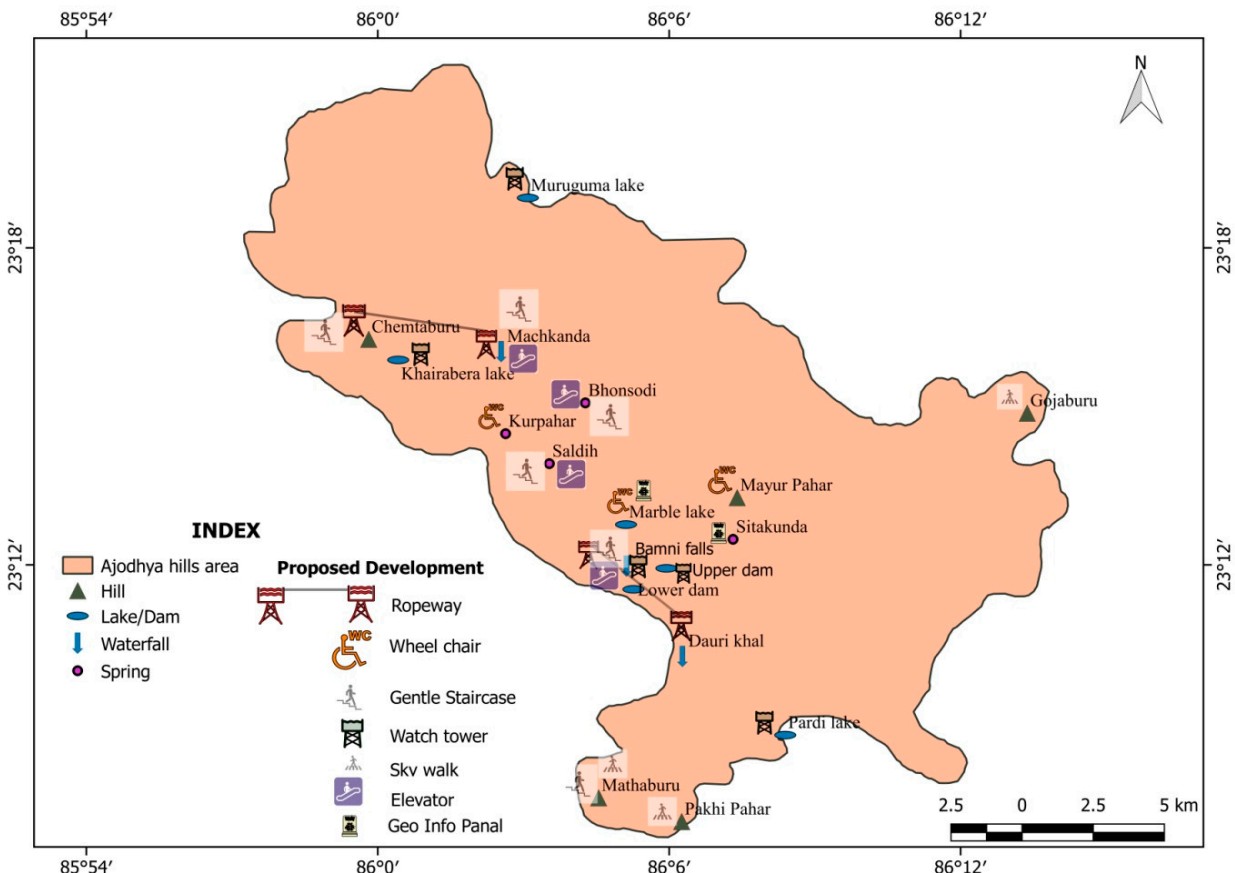

**Figure 7.** Proposed site-specific planning measures for inclusive tourism development in the Ajodhya hills.

## 5. Conclusions

The introduction of inclusive tourism may be a tool to spread economic benefits to areas not frequented by international tourists [36,37]. In spite of huge geotourism resources,

the study area has not received a considerable number of foreign visitors to date. With the advent of inclusive tourism, tourists with special needs, along with other tourists, could be given the opportunity to fully enjoy their holidays [9]. For sustainable geotourism in areas such as the Ajodhya hills, the primary requirement is to ensure accessibility to the geosites, most of which are located in inaccessible terrain. So, in making attraction hubs accessible to all people irrespective of their physical barriers, there are huge opportunities in the field of inclusive tourism development to be realized [38]. Thus, inclusive tourism not only serves people with special needs; it also extends opportunities for those travelers who consider safety to be a priority [39]. Inclusive tourism ideologically aims to create an environment where people of all abilities are welcomed as esteemed customers and guests, and it has the long-term agenda of expanding employment opportunities and redistributing resources to advocate for a pro-poor tourism [40]. Inhabited mostly by the tribal people, the Ajodhya hills is a poor, backward area. Developing inclusive tourism is considered a priority for such tourism destinations as inclusion is one of the key principles behind the United Nations Sustainable Development Goals [41]. Because of attractiveness from a geotourism perspective, opportunities arise for the development of new inclusive destinations [42]. Inclusive tourism has become an emerging industry with the development of 5G technology, which generates interest among differently abled and elderly communities in various newer types of tourism, including geotourism. Through visualization of the geosites by virtue of this new technology, more and more potential tourists may develop an interest to physically visit the sites, from which the necessity of accessible tourism infrastructure arises. Through intensive field studies, this research article has attempted to address this issue for the Ajodhya hills, which is a geotourism paradise.

**Author Contributions:** Conceptualization, P.C.; methodology, A.G. and R.M.; software, R.M. and A.G.; validation, A.G. and R.M.; formal analysis, R.M. and A.G.; investigation, A.G. and R.M.; data curation, A.G. and R.M.; writing—original draft preparation, P.C., A.G. and R.M.; writing—review and editing, P.C., visualization, P.C.; project administration, P.C. All authors have read and agreed to the published version of the manuscript.

**Funding:** This research received no external funding.

**Institutional Review Board Statement:** Not applicable.

**Informed Consent Statement:** Not applicable.

**Data Availability Statement:** Not applicable.

**Conflicts of Interest:** The authors declare no conflict of interest.

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
