# Peer review of "Inclusive Tourism Adopted to Geosites: A Study in the Ajodhya Hills of West Bengal in India"

_tourismhosp, doi:10.3390/tourhosp4020020_

Round 1
Reviewer 1 Report
The article is very interesting and well-organised in explaining the area and the method.
The approach is also fascinating and well explained in the introduction and the conclusions. The only note is related to the inappropriate use of "geosite".
The concept of geosite is defined by a long history of literature and is different from a site of geo-touristic interest, and they are not synonymous.
It is ambiguous in the article whether geosites have been defined by a previous reclassification using one of the different existing methods, although I would exclude this hypothesis.
Instead, it reclassifies geotourism sites aimed at inclusive and sustainable tourism.
If the authors intend to use the term geosite, I believe a chapter should be included in which the sites of interest are reclassified to be described as geosites. Otherwise, they can be defined as sites of geological interest, elements of geodiversity, or key sites for inclusive geotourism.
Author Response
A separate paragraph is added in section 2. Study area for the fulfillment of the requirement as asked for.
Reviewer 2 Report
The article discusses the interesting topic of inclusive tourism in a selected area in West Bengal, India, which is known for its geotourism attractions. The article is attractively written, presenting interesting geosites and their potential tourist use.
The literature review is correctly written. I would only appreciate it if it used more sources from other parts of the world as well (e.g., from Europe, where there are many similar studies). The methodology is created logically and consistently; the results are presented in a clear way. In Conclusion, suggestions for improvement are drawn from the results.
A few things in the article deserve some editing:
Figure 5 is too small and illegible; it would be appropriate to move the photos below the map and enlarge the entire Figure to the width of the entire page.
Figure 6 - what about category boundaries? In the Figure, it looks like, for example, geosite 33 is on the border of MH and MM. Table 6 then states that it is in the MH category. This should be clearly visible in the Figure as well.
Author Response
- A few citations have been added to satisfy the requirement of highlighting similar studies already done in different parts of the world.
- The photos in the figure 5 have been rearranged and placed at below the map that shows area of interest.
- The justification to include border cases in the next higher order category have been explained at the bottom of table 6.
Round 2
Reviewer 1 Report
This paper is very interesting. I congratulate the authors on their writing.
Reviewer 2 Report
All issues have been addressed. The manuscript is acceptable now.